# Shaping Exploration: How Does the Constraint-Induced Movement Therapy Helps Patients Finding a New Movement Solution

**DOI:** 10.3390/jfmk8010004

**Published:** 2022-12-22

**Authors:** Matheus M. Pacheco, Luisa F. García-Salazar, Laura H. S. C. Gomes, Fabiana S. Marques, Natalia D. Pereira

**Affiliations:** 1CIFI2D, Faculty of Sport, University of Porto, 4200-450 Porto, Portugal; 2Rehabilitation Science Research Group, School of Medicine and Health Sciences, Universidad del Rosario, Bogotá 65150-000, Colombia; 3Department of Physiotherapy, Federal University of São Carlos (UFSCar), São Carlos 13565-905, Brazil; 4Motor Development Group, Department of Physical Education, Federal University of Rondônia (UNIR), Porto Velho 78900-000, Brazil

**Keywords:** rehabilitation, motor learning, dynamical systems, physiotherapy, search-strategies

## Abstract

Despite the relative success of constraint-induced movement therapy in the recovery of injury-/trauma-related populations, the mechanisms by which it promotes its results are still unknown. From a dynamical systems approach, we investigated whether the induced exploratory patterns within and between trials during an exercise in Shaping (the therapy’s practice) could shed light on this process. We analyzed data from four chronic spinal-cord injury patients during a task of placing and removing their feet from a step. We assessed the within and between trial dynamics through recurrent quantification analyses and task-space analyses, respectively. From our results, individuals found movement patterns directed to modulate foot height (to accomplish the task). Additionally, when the task was manipulated (increasing step height), individuals increased coupling and coupling variability in the ankle, hip, and knee over trials. This pattern of findings is in consonance with the idea of Shaping inducing exploration of different movements. Such exploration might be an important factor affording the positive changes observed in the literature.

## 1. Introduction

Constraint-Induced Movement Therapy (CIMT) has been relegated as one of the most effective interventions for recovering functionality in a range of injury-/trauma-related populations (e.g., stroke patients, cerebral palsy, spinal cord injury) [1]. This protocol is, in fact, a package that includes task-oriented training, adherence-enhancing behavioral strategies, and the directed use of the more-affected limb [2]. Despite several studies demonstrating its effectivity [3,4,5] or detailing its components [6], there is still a paucity of studies investigating how and why the protocol (as a whole or each of its parts) shows the observed effects.

A recent article [3] observed the gait kinematics of chronic stroke patients throughout the CIMT protocol through the lenses of the dynamical systems approach [7]. The authors found that the expected “path” of change postulated by the theory was indeed observed—which led to improvements in the gait pattern. That is, the patients showed a decrease in stability of the initially (rigid) stroke gait pattern followed by a change in the movement pattern and a later stabilization (30 days later): a phase transition. The authors argued that the intensive and directed task-oriented training (called Shaping) was sufficient to perturb the initially stable system and constrain the patient to search for better solutions.

The main idea of Shaping is to promote functional gains through a small but continuous increase in task difficulty and intensive practice (10 days of 3 h of practice). The individual receives continuous motivational support and feedback on his/her performance. Changes in the task are usually parametric changes in the distance to reach, the mass of the implement manipulated, the time to finish the movement, and the number of repetitions. These manipulations are thought to destabilize the current stable pattern, promoting search through other perceptual-motor possibilities [3].

Despite the speculation being in line with current developments of the dynamical systems approach that emphasize learning as a process of search in the perceptual-motor workspace (see [8,9]), there is no empirical demonstration of such process during Shaping that would sustain the aforementioned statements. In fact, notwithstanding the current support for the ideas on search/exploration in motor learning literature (e.g., [10,11,12,13]), there is no consensus on how to grasp this exploratory process that would, according to these recent theories, be the leading process for behavioral change.

Therefore, in the present exploratory study, we investigated the changes occurring in an exercise in Shaping (the practice stage of CIMT). Our goal here is not so much to demonstrate how exploration occurs during the exercise; rather, we aim to observe how two different “approaches” (task space [9,14] and recurrent quantification analyses [15]) capture the exploratory behavior during the session. These two approaches emphasize differently between and within trial dynamics, respectively, and, thus, might provide different insights into the process.

## 2. Materials and Methods

This study was approved by the Human Research Ethics Committee of the Federal University of São Carlos (Report 25081219.2.0000.5504), in accordance with the Declaration of Helsinki and the National Health Council Resolution 466/2012. All procedures were explained to the participants before they signed the informed consent form. The study occurred at the Multidisciplinary Center of Movement Analysis of the Federal University of São Carlos (UFSCar, São Carlos, Brazil).

### 2.1. Participants

A convenience sample of 4 participants with chronic spinal cord injury (lesion time > 6 months) participated in this study. Participants had the ability to walk eight meters with or without an assistive device. None of the participants were currently undergoing another rehabilitation treatment and did not have another neurological disease. Table 1 presents detailed clinical and demographic characteristics.

### 2.2. Procedures

Participants performed a shaping task that consisted of placing and removing the foot of the participant’s preferred lower limb from a step as quickly as possible. A stepper was placed at a 15 cm distance from participants’ feet. The height was determined considering each participant’s capabilities and perception to perform a challenging task using the Rating of Perceived Challenge (RCP) scale. This scale ranges from 0 to 10, where 0 means not challenging at all; and ten means the most challenging task ever performed by the participant. The experimenter modified the height of the stepper until the patient gave a score between 6 and 8; the resultant height was defined as the stepper’s height to perform the task. Upper extremity support on a walker or a cane of the participants was allowed.

The task was divided into two blocks of five trials with ten repetitions each, for a total of 100 repetitions. After the first block (50 repetitions), the experimenter provided feedback about the time spent in the first and last trials to the participant. Also, the experimenter reapplied the RCP scale to guarantee that the task was still challenging. If the score was lower than the RCP provided in the first block, an adjustment in the height of the stepper was made until the RCP reached a score between 6 and 8. In this study, all four participants had their height increased for the second block. After that, the second block started.

The movements during the shaping task were collected using seven cameras of the ProReflex Motion Capture System (Qualisys Inc., Gothenburg, Sweden) at a sampling frequency of 120 Hz. Reflexive passive markers were positioned (bilaterally) on the first, second and fifth metatarsal heads, medial and lateral malleoli, medial and lateral epicondyle of the femur, major trochanters, iliac crests, acromion, C7, and sternum. Additionally, clusters were positioned at T4, T10, and bilaterally on the lateral faces of the thighs and legs. Afterwards, a static anatomical position trial was conducted, and then participants started to perform the shaping task as previously described.

The joint angular measures were processed in the Visual 3D™ software (v4., C-Motion, Inc., Rockville, MD, USA), considering the joint coordinate system recommended by the International Society of Biomechanics [16,17], and were expressed relative to the static anatomical position.

### 2.3. Data Analysis

All necessary data and codes are available online at OSF (https://osf.io/3n4tk/—accessed on 22 November 2022).

First, to guarantee that we are referring to a practice situation, in that we found improvement in functionality, we also report the Lower-Extremity Motor Activity Log (LE-MAL) before and after the overall intervention (see [18]). LE-MAL is a semi-structured interview that consists of questions about the participant’s level of assistance, performance, and confidence while executing 14 different daily living activities. The result is composed of assistance, functional performance, and confidence scales. The arithmetic mean of these three scales is the composite score.

Provided technical and logistic issues, the third block of ten trials of participant two and five blocks (1st, 2nd, 4th, 5th, and 6th) of participant 4 could not be analyzed. The kinematic data were filtered with a low-pass Butterworth filter (fourth-order, zero-lag) at 6 Hz. Furthermore, the raw data on the displacement of the foot and sacrum, and their anatomical position were exported.

To extract each trial per block, we designed a specific algorithm to identify the beginning and end of each “cycle” of movement. First, we found the first moment that the time series surpassed the mean height value for the first ten frames added of ten times the standard deviation. The ten standard deviation values were to guarantee that no initial small motion (e.g., repositioning of the body) would be identified as a trial. Furthermore, we identified the moment between steps by finding the minimum height of the foot position between peaks. For this, using the findpeaks function in Matlab (R2020b), we found the maximum of the time series (multiplied by −1) within a window of 200 frames with a value at least bigger than the average of the time series. To guarantee that the positions were correct, the experimenter inspected each block of ten trials visually, and, if not, the experimenter selected manually the correct moments. Consequently, with the minima annotated, the time series was cut into cycles for further analysis. The cycles were all normalized to have a 1-to-100%-time range. The actual time per step was also noted.

A similar algorithm was designed to identify the moment at which the foot reached its peak height before the step. For this, using the findpeaks function in Matlab, we found the maxima of the time series within 200 frames with a value at least bigger than the time series mean. To guarantee that the ten peaks were indeed the peak foot height, the experimenter visually inspected each block of ten trials, and, if not, the experimenter selected manually the correct moments.

#### 2.3.1. Within-Trial Exploration

To understand the within-trial exploration, we utilized recurrence quantification analyses (RQA) [19]. The main idea of these analyses is that by observing (and quantifying aspects of) the recurrent patterns of a given time series, one can understand features of the underlying dynamics that give rise to the observed data. A simple hypothetical example comes from comparing how the knee flexion/extension angle time series recur (show similar values) in different terrains: on a regular flat surface, the angle values would recur quite periodically; on an irregular surface, the angle values would still recur, but now in a more irregular pattern, demonstrating that the interaction between body and soil is now different from the first situation. Still, in both cases, recurrence would not demonstrate randomness.

For our case, imagine a situation that an individual is “trapped” in a non-functional (but highly stable) movement pattern. This individual would show a highly periodic movement pattern with lower entropy, high stability, and a deterministic structure. Now, if the task introduced during Shaping pushes the system to a less stable regime, we will see a higher value of entropy, decreased stability, and a deterministic structure. Our question is not whether a perturbation modifies the observed dynamics as this is the definition of a perturbation and the difficulty level (through RCP) is expected to be sufficiently “perturbing”. The question is what evolves from it. If the task requires that the patient finds novel solutions (explore), the system might increase entropy even further. Nonetheless, if the system finds a solution soon or “recovers” from the perturbation returning to the initial state, the system might return to previous values of stability. Furthermore, a new increase in step height occurs and a new loop of these processes might occur.

Taking advantage of all the measures that can be extracted through RQA, we investigated how the Shaping intervention leads the system to a new movement pattern (as demonstrated in [3]). For this, we performed two RQA analyses that take advantage of all dimensions (lower limb joint motions) of the task and one RQA analysis that uses only knee and hip angles (flexion/extension), as these were the most involved joints in the task (see the results).

First, we investigated how the system recurred in terms of all its dimensions through two analyses: an auto RQA after dimension compression through principal component analysis (“PCA-aRQA”) and a multidimensional RQA (mdRQA). For the PCA-aRQA, we first performed a principal component analysis (covariance-based) using all dimensions from the hip, knee, and ankle joints (7 dimensions) and selected the first component (the one that takes most of the variance of the data set). We noticed that the results of the PCA (for each block of ten trials, independent of height) were similar over time (quantified using the dot product of the first eigenvector). Thus, we understood that the system was not changing the main joints for the task. Thus, using the eigenvalue time series of the first component, we performed an aRQA to investigate how the system changed over trials in the main mode of coordination in this task. We followed the steps of ref. [19] to define the parameters of RQA (the same procedures were utilized for the subsequent recurrence quantification analyses).

Second, observing that knee and hip (extension/flexion) were the time series that most participated in the observed action, and we also performed a cross RQA (cRQA) using these two time series only. The cRQA sees a pattern of cooccurrence between two time series—when a given value (usually from a normalized time series varying in magnitude from 0 to 1) also occurs in a different time series. Theoretically, one can run RQA in as many dimensions as required. Thus, in order to observe whether we can extract more information when considering all dimensions, we performed the mdRQA [20]. The embedding dimension and delay were set to zero for mdRQA, since the set of joints has seven degrees of freedom, and the embedding dimension for a single dimension (using a false nearest neighbor algorithm) was around eight.

For all three RQA analyses, the measures utilized for discussion were recurrence, entropy, determinism, LMAX, and laminarity. Recurrence measures the percentage of recurrent instances. Determinism measures the percentage of recurrent points followed by another recurrent point. LMAX measures the length of the longest recurrence trend in time. Entropy measures the distribution of the lengths recurrence trend in time (Shannon information entropy of all diagonal lines in the recurrence plot). Laminarity measures the tendency of the time series to stay within a single state.

#### 2.3.2. Between-Trial Exploration

The analysis of the within-trial dynamics has the issue of emphasizing “equally” the movement trajectory to reach the goal and the “reaching-the-goal-moment” (or the task-relevant moment of the task). Note that these two “categories” have different properties; the trajectories can vary, maintaining the relevant moment [20], or the trajectories can be quite similar with small adjustments in the relevant moment [21]. Thus, in order to also study how individuals explored a relevant moment of the task, we analyzed how (in terms of the hip and knee joints) the participants reached the highest height with their feet when attempting to place their feet on the step.

For this, we used what some have called “task-space analysis” (see [9,14]). Different from the RQA, which has specific procedures and measures, the task space analysis is rather a graphical illustration of how “movement variables” relate to the movement outcome (for a thorough discussion on selecting these, see “elemental variables” in [22]). The main idea, nonetheless, is to understand how individuals coordinated redundant degrees of freedom to reach a given performance or outcome. Many analyses are based on this idea [23,24,25], and we will be using variants of these.

First, using the hip and knee at the foot peak height for each trial, we calculated the area of the ellipse (encompassing 85% of the data) in this two-dimensional space. Second, we explored whether the individual explored the space in terms of equivalent solutions (new movement patterns that lead to the same outcome) or explored novel solutions that would decrease or increase the foot peak height (orthogonal to the equivalent solutions). For this, we considered a two-dimensional model of the hip and knee resulting in the observed heel height:Heel_z_ = hip_z_ + thigh_l_ ∗ sin (hip_a_) + shank_l_ ∗ sin (knee_a_ + hip_a_)(1)
where heel_z_ is the heel position in the z-axis (vertical), hip_z_ is the hip position in the z-axis, thigh_l_ is the length of the thigh, hip_a_ is the hip angle (relative to the horizontal axis), shank_l_ is the length of the shank, and knee_a_ is the knee angle (relative to the thigh). This model tended to underestimate the actual peak foot height by 10.52 ± 18.09 cm, provided we did not include all other dimensions in the model. Furthermore, using the moment of foot peak height as our basis, we followed ref. [24] and calculated the equivalent and orthogonal spaces of the mean peak height. The former describes the combinations of the knee and hip angles that lead to no changes in peak height and the latter the combinations that change peak height. The data were projected to each space, and the variance of each was calculated per block of movements.

Third, we calculated the tolerance cost of each block of movements following ref. [26]. Knowing the non-linearity of the relations between movement patterns and a given peak height, the same variance around different mean movement configurations (e.g., knee and hip relations) can lead to different variances in the movement outcome (foot peak height). To understand whether individuals were exploring in terms of finding mean movement solutions more tolerant to variability, we calculated the tolerance cost of each block through the TNC approach [23,26].

Fourth, to understand the dynamics of exploration between trials, we performed the analysis described in refs. [27,28]. This analysis, first, finds points of discontinuity in the movement variables (in the present case, the knee and hip angles at the foot peak height) and trial-to-trial (linear) trends. Second, after identifying these discontinuities with the movement variables, the algorithm evaluates whether the sections between discontinuities present trends that demonstrate maintenance (no change in either movement variables or movement outcome), covariation (change in the movement variables that maintain movement outcome unchanged), or change (change in the movement variables that lead to changes in movement outcome). Furthermore, the discontinuities are classified as “jumps” or “no-jumps”; the first means that the mean value of movement variables before the discontinuity is changed after the discontinuity.

### 2.4. Statistical Analyses

For the overall measures, within and between-trial exploration, we fitted the data with a linear mixed-effect model for each variable considering the step height and blocks (and their interaction) as independent variables. We only considered the constant as a random measure. For all measures (except trial-to-trial dynamics), we used the fitlme code in Matlab. For the trial-to-trial dynamics, we used the fitglme code considering a binomial distribution. For all analyses, we considered a value of alpha of 0.050.

We are aware of the lack of power in our analyses given the small sample size. We performed these analyses to guide discussion and avoid biased interpretations of descriptive statistics. We emphasize that this is an exploratory study. All the data were analyzed in Matlab R2020b.

## 3. Results

### 3.1. Performance Measures

To guarantee that the current practice was related to a successful increase in functionality, we describe the results of LE-MAL. All participants improved their LE-MAL scores. Participant 1 increased from 5.36 to 7.88, participant 2 from 6.04 to 7.01, participant 3 from 4.08 to 6.72, and participant 4 from 5.98 to 6.66.

Figure 1 shows the feet height and the knee and hip angle-angle plot during ten movements of an exemplary participant (Participant 1). In Figure 1a, we can see two small peaks for each cycle, the first represented the feet height to step and the second the increased height to return the feet to the ground. Figure 1b shows that the coordination of this individual demonstrated a proportional flexion of the hip and knee (increase and decrease of degrees, respectively), with large knee participation towards the end. This pattern was common for all four participants—not considering potential peculiarities (discussed below).

The tables with all performance measures are presented in the Appendix A. The linear mixed effect model showed, for clearing height (Figure 1c, Appendix A), a main effect of step height (*β* = −0.017; *t* (30) = 2.48; *p* = 0.018); no other effect was significant (*p*’s > 0.050) (*R*^2^ = 0.93). This shows a decrease from 7.9 cm (in the first step height) to −1.7 cm in the second step height. Provided this decrease was smaller than the increase in actual step height, this represents an increase in foot peak height (Appendix A). Indeed, we found a significant main effect for step height (*β* = 0.022; *t* (30) = 3.49; *p* = 0.002); no other effect was significant (*p*’s > 0.050) (*R*^2^ = 0.99).

Considering the joint angles at foot peak height (Figure 1d–f, Appendix A), we found, for hip angle, no significant effects (*p*’s > 0.050). For knee angle, we also found no significant effects (*p*’s > 0.050). For ankle angle, we found a significant effect of blocks (*β* = −0.42; *t* (30) = 2.70; *p* = 0.011); no other effect was significant (*p*’s > 0.050) (*R*^2^ = 0.99).

For joints range of motion (Figure 1g–i, Appendix A), we found, for the hip angle, no significant effects (*p*’s > 0.050). For knee angle, we also found no significant effects (*p*’s > 0.050). For ankle angle, we also found no significant effects (*p*’s > 0.050).

For time spent per block (the sum of the ten movements), we did not find any significant effect (*p*’s > 0.050).

Considering the coordination between limbs during trials, Figure 2 shows the average loading of each joint in the first component from the principal component analysis considering all joints for both step heights (first column) and the projected hip and knee flexion time series in this first component (second column). The loadings are similar between the first and second step heights (normalized dot products between step heights around 0.99).

### 3.2. Within-Trial Exploration

Figure 3 shows the auto and cross recurrence plots and their respective output measures (including the multidimensional recurrence quantification analysis output). We organized this section in terms of outcome measures to understand the within-trial exploration as a function of the input measures. The tables with all RQA measures and linear mixed effect models’ results are presented in the Appendix A.

#### 3.2.1. Recurrence

Considering the aRQA, we found a significant interaction effect between blocks and step height (*β* = 0.17; *t* (30) = 2.13; *p* = 0.041); no other effect was significant (*p*’s > 0.050). It means that individuals, after an increase in step height, increased their recurrence over blocks. For cRQA, we failed to find any significant effect (*p*’s > 0.050). For mdRQA, we found a significant main effect of step height (*β* = 0.65; *t* (30) = 2.62; *p* = 0.014) and an interaction between step height and blocks (*β* = −0.17; *t* (30) = 2.24; *p* = 0.032); block was not significant (*p* = 0.975). It means that there was an increase in recurrence after the step height change but a subsequent decrease in recurrence over blocks.

#### 3.2.2. Determinism

For aRQA, we found no significant effects (*p*’s > 0.050). For cRQA, we also found no significant effects (*p*’s > 0.050). For mdRQA, we also found no significant effects (*p*’s > 0.050).

#### 3.2.3. Entropy

Considering the aRQA, we found no significant effects (*p*’s > 0.050). For cRQA, we found no effects (*p*’s > 0.050). For mdRQA, we found an effect of step height (*β* = 0.33; *t* (30) = 2.42; *p* = 0.022) and an interaction between height and blocks (*β* = −0.10; *t* (30) = 2.37; *p* = 0.024); there was no main effect of block (*p* = 0.727) (*R*^2^ = 0.87). It means that there was an increase and a subsequent decrease in entropy over blocks when the second step height was introduced.

#### 3.2.4. LMAX

For aRQA, we found no significant effects (*p*’s > 0.050). For cRQA, we also found no significant effects (*p*’s > 0.050). For mdRQA, we found the same pattern of results as in the entropy and determinism measures. We found a significant main effect for step height (*β* = 3.06; *t* (30) = 2.94; *p* = 0.006) and an interaction between step height and blocks (*β* = −0.83; *t* (30) = 2.64; *p* = 0.013); there was no main effect of blocks (*p* = 0.421) (*R*^2^ = 0.87). It means that there was an increase and a subsequent decrease in LMAX over blocks when the second step height was introduced.

#### 3.2.5. Laminarity

For aRQA, we found no significant effects (*p*’s > 0.050). For cRQA, we found significant main effects of step height (*β* = −16.35; *t* (30) = 2.70; *p* = 0.011) and blocks (*β* = −3.23; *t* (30) = 2.43; *p* = 0.021); the interaction effect was not significant (*p* = 0.181). It means that there was a continuous decrease in laminarity over blocks from the first to the second step height. For mdRQA, we found no significant effects (*p*’s > 0.050).

### 3.3. Between-Trial Exploration

For the between-trial exploration, we investigated the overall dispersion pattern of the trials (through general variance, variance along the equivalent and orthogonal space and analysis of tolerance) and the trial-to-trial dynamics. Figure 4 shows the relation between the knee and hip (flexion) at peak foot height for each participant for all movements and the different step heights.

#### 3.3.1. Dispersion Analysis

Figure 5 shows the data for each of the dispersion variables. The overall variance across blocks and step height did not show change (*p*’s > 0.050).

Considering variance in equivalent and orthogonal spaces, for the former, we found no main effects of blocks, step height, or their interaction (*p*’s > 0.050). For the latter, we found a significant main effect of blocks (*β* = 5.20; *t* (30) = 2.88; *p* = 0.007), demonstrating an increase in variance along the orthogonal space. The other effects were non-significant (*p*’s > 0.050).

Finally, considering whether individuals moved to more or less tolerant spaces in the task space, we found a significant effect of blocks (*β* = 0.0033; *t* (30) = 2.72; *p* = 0.011), demonstrating an increase in tolerance cost over blocks. The other effects were non-significant (*p*’s > 0.050).

#### 3.3.2. Trial-to-Trial Dynamics

Figure 6 shows the trial-to-trial dynamics of participant 4. The participant did not show any instance of continuous change (only “maintenance”) or an instance of a jump. This pattern of maintenance and jumps was similar for all participants.

In terms of jumps, we found a 50% chance of the occurrence of jumps during practice. The general linear mixed model showed no effects of step height, blocks, or their interaction (*p*’s > 0.050).

## 4. Discussion

The present study attempted to understand how the practice of one exercise of Shaping (part of CIMT) would modify how individuals explore new movement possibilities within and between trials through the lenses of the dynamical systems approach to motor behavior. Overall, the results show that step height induced changes in the within-trial exploration patterns but did not affect the ongoing change observed through blocks in the between-trial exploration pattern. More specifically, recurrence, entropy, and LMAX (of mdRQA) of the within-trial dynamics increased after the change in step height with a later resettlement into values of the first step height. In the between-trial dynamics, individuals showed a preference to explore movement patterns that would modify step height, finding less tolerant regions through discontinuous changes in the mean movement pattern configuration. All this while demonstrating increased step height and functional improvement (at the end of all interventions).

Before delving into the interpretation of the within and between-trial dynamics, we acknowledge that the observed change in peak foot height is not expressive. However, the fact that individuals improved in such a brief period of practice is also noteworthy. Furthermore, we observed that while the participants were still able to increase foot peak height, they decreased the clearing height. This demonstrates that they were reaching their maximum capacity; the task was challenging enough for their “clearing margin” to decrease.

Maintaining performance close to the participant’s limits seems to be a defining feature of this protocol [2,29]—an argument for its efficacy. Importantly, the possibility of maintaining this level of difficulty was only possible through the use of the RCP [30]—which is not common even in CIMT. Usually, one uses a standard value to modify the task requirements (increasing difficulty); this can over or undershoot actual individual capabilities. The idea to monitor, through RCP, when to increase difficulty seems suited.

The within-trial analyses demonstrated that the increase in step height affected entropy, recurrence, and the LMAX of the multidimensional RQA. Following ref. [31], the recurrence analysis of more than one variable considers how the coupling of these variables presents itself over time. From the increase in LMAX, we know that the maximum period of coupling between variables increased when the experimenter modified step height. Such an increase in the coupling period occurred often considering that recurrence also increased. Noticeably, the increase in coupling was not homogeneous as their dispersion also increased (as denoted by the increase in entropy).

Considering the similar dynamics of the three measures, we infer that a similar cause is in place. We infer that individuals attempted to modify their coupling in order to deal with the new task constraints. This exploration along with coupling possibilities—exploring different movement patterns—would increase (similarly) recurrence, entropy, and LMAX. Logically, this would leave laminarity and determinism unchanged: the amount of determinism was already close to the maximum and there is no need for increased maintenance in a given state for longer periods.

Other studies that employed RQA did not observe this pattern of results. This might have occurred since the studies do not report the same outcome variables as the present study or they have focused on differentiating patients from healthy controls. For instance, Labini et al. [32] demonstrated that hypovestibular subjects show lower recurrence and determinism compared with healthy controls in gait. In the same vein (but with a distinct pattern of results), Parkinson’s patients tend to show higher entropy, determinism, and divergence (lower LMAX) compared with healthy controls [33] in postural control. One of the few studies concerned with changes within an individual [15] showed that healthy individuals—in learning to judge weight or width through wielding objects—decrease entropy values over time.

It is important to emphasize that we observed results from step height manipulation mainly through mdRQA. We can highlight the fact that mdRQA is the only analysis performed that encompassed all dimensions of the movement—being more sensitive to changes at this stage of the therapy. We also found trends for aRQA (recurrence increased over blocks in the highest step height) and cRQA (laminarity showed a continuous decrease over blocks). Nonetheless, provided the small sample size and the lack of other systematic trends (as observed for mdRQA), we refrain from speculating on the observed changes in these two recurrence quantification analyses.

The between-trial analysis demonstrated that, if we are correct that within-trial dynamics referred to exploration, this exploration was in terms of finding movement patterns increasingly in line with foot height variance. That is, individuals were searching for movement patterns that tend to vary modulating foot height. Exploration for novel solutions (ones that increase step height) will inevitably be along the orthogonal and need not be directed to increased tolerance. Such a pattern in the results is contrary to most studies on task space analyses, as all of them are concerned with matters of maintaining a given value (e.g., reaching the same point in space, maintaining the center of mass above the base of support, throwing to a target) rather than maximizing it (e.g., [25,26,34,35]). However, these results are in line with the present task requirements.

Additionally, in line with previous studies in complex movements, the trial-to-trial exploration follows a maintenance/jumps pattern. The same result was observed in Pacheco et al. [28] analysis of throwing in healthy participants. This indicates that continuous exploration of the perceptual-motor workspace is not as common as studies based on simple tasks claim (e.g., [27,36,37]).

Considering this pattern of results, we support the idea that Shaping—through increased difficulty controlled by the RCP—leads to an increased exploration of movement patterns (increased entropy in the system) that later settles into a “sufficient” solution suited to the task at hand (in this case, modulate foot height). This follows the idea that the system is searching through task space and perceptual-motor workspace [9]. As one (the experimenter/therapist) manipulates the task space—requiring higher step heights—individuals must leave the current perceptual-motor workspace region. This inevitably creates a moment of uncertainty, as the relationship between perceptual-motor workspace and task space changes requiring further exploration.

Clearly, there is the possibility that foot peak height is a control parameter that leads to instability with the movement pattern (old attractor state), with later decay to the new attractor state [38,39]. This would also explain the aforementioned results. In fact, this is not contradictory to the explanation above—it is a higher-level description of our explanation. Nonetheless, we did not find evidence of new movement dynamics (no differences in PCA of the first and second step height). Thus, it is less tempting to argue for a bifurcation (change in the attractor layout given control parameter changes). Following the search idea, individuals might have explored within a stable coordination mode (as demonstrated in ref. [40] and postulated in ref. [41]).

A feature that the search approach allows that might be fortuitous to understanding the change in CIMT (and Shaping) is that it considers the individuality in the perceptual-motor workspace (see [42,43], common to the dynamical systems approach as well [38,44]). Observing Figure 1d–f, we find that individuals explored their hip, knee, and ankle positions differently to deal with the task. Such differences could not be explored here as the number of participants is small. However, it is clear that this venue should be considered when understanding, for instance, why participant 3 could not maintain his clearing height when the step height was increased while participant 4 could.

We acknowledge the limitations in the present exploratory study provided the sample size. Nonetheless, it is always difficult to recruit and perform interventions in a large sample of patients (of any kind). The generalization of the current results must be made with caution, and we invite others to replicate our results. We indicate that confirmatory studies must address the within-trial exploration results—as they are the main findings (i.e., entropy, recurrence, and LMAX). Furthermore, the within/between differentiation made here is arbitrary. For the recurrence quantification analyses, we compared in a between-trial analysis the within pattern demonstrated—which is clearly not a within-trial concern. The task space analyses carry a single point of the trial that is important but is also a result of how the individual is exploring within each step cycle. Patients/learners discover relevant/irrelevant dimensions of the task within and between trials.

## 5. Conclusions

Constraint-Induced Movement Therapy presents itself as an effective intervention in recovering patients’ functional capacity in their activities of daily living. Despite its positive outcomes, studies are necessary for one to understand its process. The current study described how behavior changes in a typical exercise included in the CIMT practice (Shaping). We found that the practice continuously challenges individuals at the limit of their capabilities, while they demonstrate small, but positive, increments in performance. Furthermore, we found that when the challenge is increased, individuals demonstrate signs of increased exploration—in the present task, through modulation of coupling between joints. This finding is in line with the dynamical systems approach to motor behavior and, more specifically, to the perspective that individuals, in learning/or re-learning, search through the space of movement possibilities [3,9].

## Figures and Tables

**Figure 1 jfmk-08-00004-f001:**
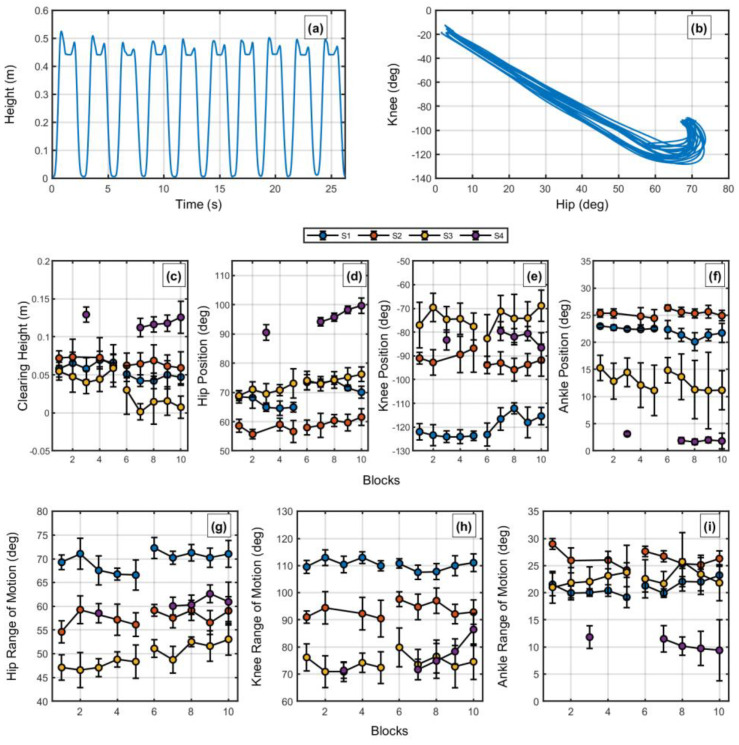
(**a**) Foot height over ten stepping movements in the task of an exemplary participant. (**b**) Angle-angle plot of the same exemplary participant for knee and hip flexion over ten stepping movements. (**c**) Average clearing height (peak foot height minus the step height) of all participants per block (ten movements) and step height. (**d**) Average hip position (flexion) at peak foot height of all participants per block (ten movements) and step height. (**e**) Average knee position (flexion) at peak foot height of all participants per block (ten movements) and step height. (**f**) Average ankle position (flexion) at peak foot height of all participants per block (ten movements) and step height. (**g**) Average hip range of motion of all participants per block (ten movements) and step height. (**h**) Average knee range of motion of all participants per block (ten movements) and step height. (**i**) Average ankle range of motion of all participants per block (ten movements) and step height.

**Figure 2 jfmk-08-00004-f002:**
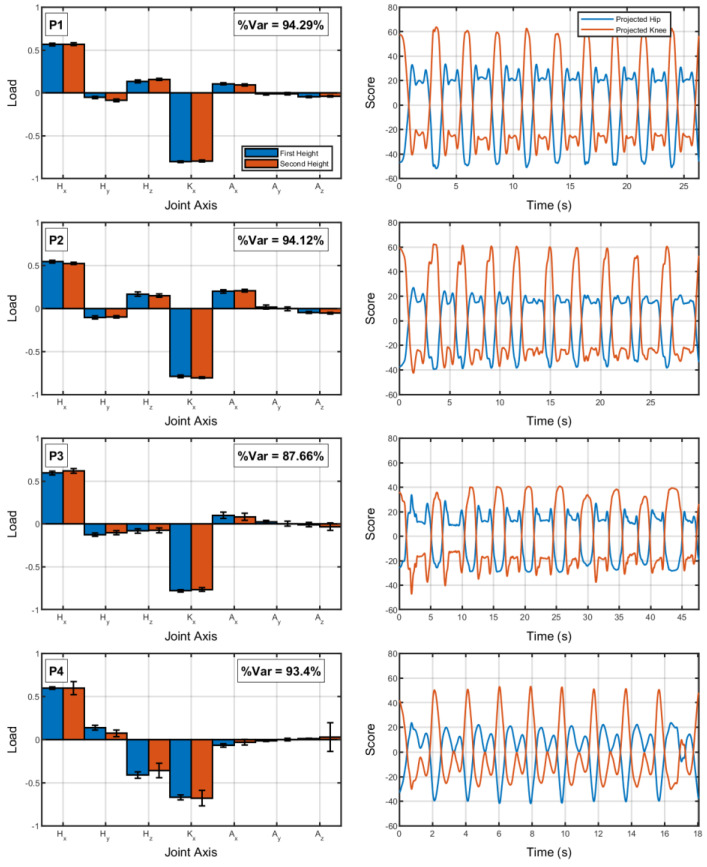
First column: Loadings (and percent variance accounted for) of the first component of the principal component analyses performed on hip, knee, and ankle joints (all seven dimensions). (**P1**–**P4**) refer to each participant. Second column: Projection of hip and knee (flexion-extension) on the first principal component over time.

**Figure 3 jfmk-08-00004-f003:**
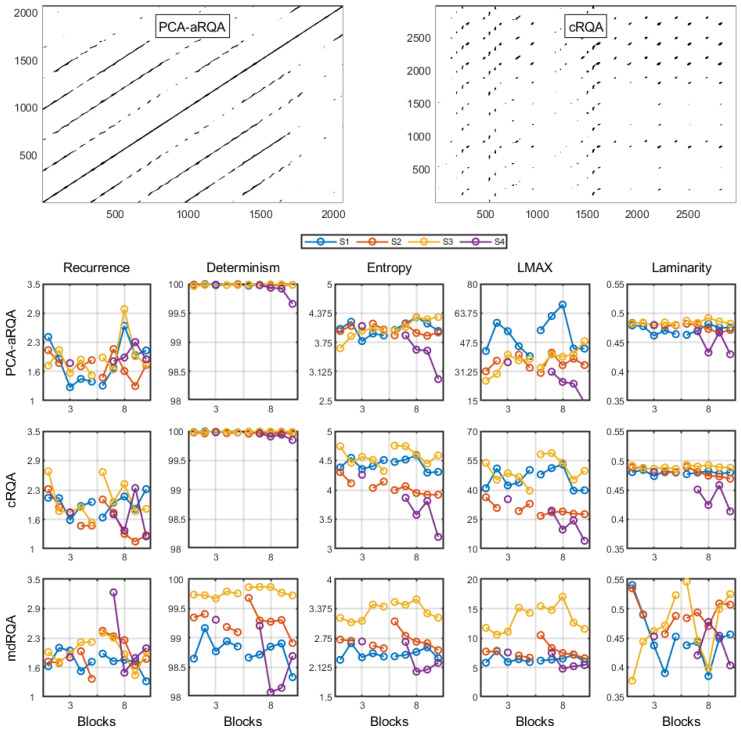
Recurrence plots (top row) and the recurrence quantification analyses (PCA-aRQA, cRQA, and mdRQA) outcomes for all four participants.

**Figure 4 jfmk-08-00004-f004:**
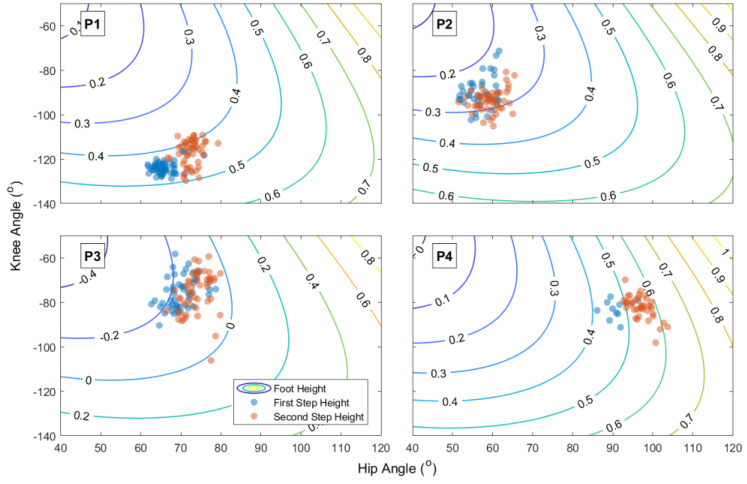
Hip/knee (flexion) relation per movement (at peak foot height) for each participant. The contour lines (and values) represent the foot height (in m). (**P1**–**P4**) refer to each participant.

**Figure 5 jfmk-08-00004-f005:**
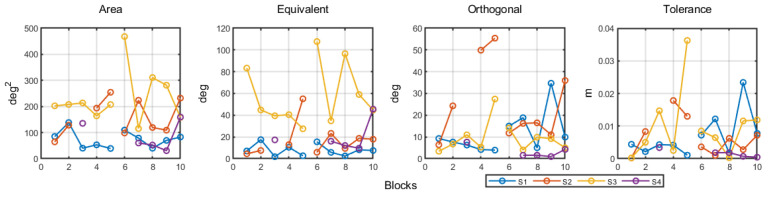
Area, equivalent variance, orthogonal variance, and tolerance cost for all participants over blocks and step height.

**Figure 6 jfmk-08-00004-f006:**
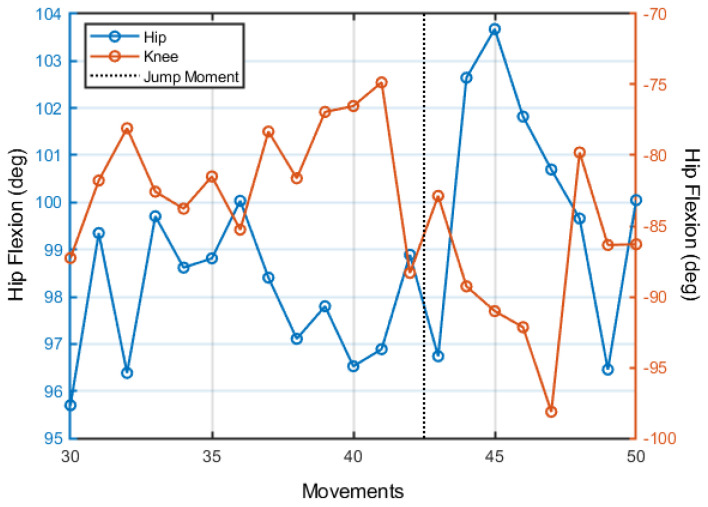
Trial-to-trial dynamics in terms of the hip and knee flexion position (at foot peak height) of participant 4. The black dashed line shows an instance of jump (discontinuous change in the trial-to-trial mean).

**Table 1 jfmk-08-00004-t001:** Clinical and demographic characteristics of the sample ^1^.

Characteristics	
Age (years)	43.2 (35–57)
Sex, male (*n*)	4
Lesion type (*n*)	
Non-traumatic	2
Traumatic	2
Time since lesion (months)	66 (12–180)
ASIA Classification	
A	
B	
C	1
D	3
E	
Neurological level	
Cervical (C2-C3)	2
Thoracic (T11-T12)	2
Lumbar	
Ankle-foot orthosis (*n*)	1
Assistive Device (*n*)	
Single point cane	2
Walker	2

^1^ Data expressed in mean and (minimum-maximum); *n* = number of participants; ASIA = American Spinal Injury Association impairment scale.

## Data Availability

All the data is available at https://osf.io/3n4tk/ accessed on 22 November 2022.

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
