# Peer review of "Shaping Exploration: How Does the Constraint-Induced Movement Therapy Helps Patients Finding a New Movement Solution"

_jfmk, 2022, doi:10.3390/jfmk8010004_

Round 1

Reviewer 1 Report

Manuscript ID: jfmk-2085210-peer-review-v1

Manuscript title: Shaping exploration: How does the Contraint-Induced Movement Therapy helps patients finding a new movement solution

Comments

This manuscript reports a case series study designed to explore the Shaping process within Constraint-Induced Movement Therapy (CIMT) using recurrent quantification and task space analyses to capture the within- and between-trial exploratory behavior. The Introduction section poses a concise background and rationale of the study. The study aims are clear and this exploratory, case report design seems adequate to address the research questions. Signal processing and analysis are well described and seem adequate to the study's aims. I have only minor suggestions for the authors to consider.

Major comments

1. Methods. Given the reported study design, I suggest the authors to adhere to proper reporting guidelines available at the EQUATOR Network (e.g., https://www.equator-network.org/reporting-guidelines/care/).

2. Materials and Methods, line 66. The National Health Council Resolution 196/96 was revoked by a new resolution 466/2012. The latter one should be mentioned given the provided report number (25081219.2.0000.5504) your research protocol was approved in 2021 by the Ethics Committee.

3. Discussion. Given the study's exploratory nature, consider suggesting what results are worthy of subsequent confirmatory analysis in larger studies, along with suggestions to decrease missing data due to technical issues as reported.

Minor comments

1. Title, lines 2-3. Possible typo (‘Contraint’).

2. Table 1. Variable levels with zeroed values may be dropped (e.g., ASIA scale A, B, E; neurologic level lumbar).

Author Response

Reviewer 1

Comments

This manuscript reports a case series study designed to explore the Shaping process within Constraint-Induced Movement Therapy (CIMT) using recurrent quantification and task space analyses to capture the within- and between-trial exploratory behavior. The Introduction section poses a concise background and rationale of the study. The study aims are clear and this exploratory, case report design seems adequate to address the research questions. Signal processing and analysis are well described and seem adequate to the study's aims. I have only minor suggestions for the authors to consider.

Major comments

  1. Methods. Given the reported study design, I suggest the authors to adhere to proper reporting guidelines available at the EQUATOR Network (e.g., https://www.equator-network.org/reporting-guidelines/care/).

Authors’ Response (AR): We thank the reviewer for the suggestion. However, our study might not fit to the CARE scope. The CARE guidelines are directed to report effects in case studies in a given intervention; our concern was with a small part of CIMT (not the whole therapy) and not much in terms of its effect but with the process of change. We modified the study objectives to avoid confusion (p. 2, l. 59-62):

“Therefore, in the present exploratory study, we investigated the changes occurring in an exercise in Shaping (the practice stage of CIMT). Our goal here is not so much to demonstrate how exploration occurs during the exercise; rather we aim to observe how two different “approaches…”

  1. Materials and Methods, line 66. The National Health Council Resolution 196/96 was revoked by a new resolution 466/2012. The latter one should be mentioned given the provided report number (25081219.2.0000.5504) your research protocol was approved in 2021 by the Ethics Committee.

AR: We modified the text accordingly (p. 2, l. 69).

  1. Discussion. Given the study's exploratory nature, consider suggesting what results are worthy of subsequent confirmatory analysis in larger studies, along with suggestions to decrease missing data due to technical issues as reported.

AR: We added a passage indicating the results that must be considered in confirmatory analysis (p. 14, l. 536-538).

“Finally, provided the exploratory nature of the study, we indicate that the within-trial exploration results are the main findings (i.e., entropy, recurrence, and LMAX) that must be addressed in confirmatory analyses.”

Minor comments

  1. Title, lines 2-3. Possible typo (‘Contraint’).

AR: The writing was corrected.

  1. Table 1. Variable levels with zeroed values may be dropped (e.g., ASIA scale A, B, E; neurologic level lumbar).

AR: We modified the table as suggested.

Reviewer 2 Report

I would like to thank the authors for the research titled Shaping exploration: How does the Contraint-Induced Movement Therapy helps patients finding a new movement solution

This article provides important information about patients with chronic spinal cord injuries. However, there are some aspects that should be corrected for its acceptance in this journal.

Abstract:

It is recommended to introduce in the Abstract information about results (data) and conclusions obtained.

Introduction:

This section is well structured and organized.

Material and Methods:

This section clearly reflects how to replicate the study, if necessary. Nevertheless, the subsection 2.4 Statistical Analyses needs to be further developed. More information is needed.

Results:

It is necessary to use the same criteria when presenting statistically significant results (p < 0.05).

Discussion:

Well founded and structured despite all the results presented.

Conclusions:

It is necessary to introduce this section to summarize the findings found in this research.

It is recommended to apply these recommendations to improve the quality of the manuscript.

Many thanks.

Author Response

Reviewer 2

I would like to thank the authors for the research titled Shaping exploration: How does the Contraint-Induced Movement Therapy helps patients finding a new movement solution

This article provides important information about patients with chronic spinal cord injuries. However, there are some aspects that should be corrected for its acceptance in this journal.

Abstract:

  1. It is recommended to introduce in the Abstract information about results (data) and conclusions obtained.

Authors' Response (AR): Overall results were added with conclusions (p. 1, l. 20-22).

We identified that individuals found movement patterns directed to modulate foot height (as to accomplish the task). Additionally, when the task was manipulated (increasing step height), individuals increased coupling and coupling variability in how ankle, hip and knee were coupled over trials.”

Introduction:

This section is well structured and organized.

Material and Methods:

  1. This section clearly reflects how to replicate the study, if necessary. Nevertheless, the subsection 2.4 Statistical Analyses needs to be further developed. More information is needed.

AR: More details were added (p. 6, l. 270-272).

“For all measures (except trial-to-trial dynamics), we used the fitlme code in Matlab. For the trial-to-trial dynamics, we used the fitglme code considering a binomial distribution. For all analyses, we considered a value of alpha of 0.050.

Results:

  1. It is necessary to use the same criteria when presenting statistically significant results (p < 0.05).

AR: Provided the need to be precise in our significant results, we maintained the style for significant results and only mentioned that the p was bigger than 0.05 (p > .050) when no significant results were found.

Discussion:

Well founded and structured despite all the results presented.

Conclusions:

  1. It is necessary to introduce this section to summarize the findings found in this research.

AR: The section was added to summarize the findings (p. 14, l. 540-550).

“The Constraint-Induced Movement Therapy presents itself as an effective intervention in recovering patients’ functional capacity in their activities of daily living. Despite its positive outcomes, much must be understood in terms of its process. The current study described how behavior is modified in a typical exercise included in the CIMT practice (Shaping). We found that individuals are continuously challenged during their practice at the limit of their capabilities, while demonstrating small, but positive, increments in their performance. Also, we found that when the challenge is increased, individuals demonstrate signs of increased exploration – in the present task, through modulation of coupling between joints. This finding is in line with the dynamical systems approach to motor behavior, and more specifically to the perspective that individuals, in learning/or re-learning, search through the space of movement possibilities (3,9).”

Round 2

Reviewer 2 Report

Thank you very much for considering my suggestions

Author Response

Dear,

We reviewed the writing to improve English writing as required by the reviewer 2.

Matheus